# Coculture Strategy for Developing *Lactobacillus paracasei* PS23 Fermented Milk with Anti-Colitis Effect

**DOI:** 10.3390/foods10102337

**Published:** 2021-09-30

**Authors:** Kai-Yi Lee, Ying-Chieh Tsai, Sheng-Yao Wang, Yen-Po Chen, Ming-Ju Chen

**Affiliations:** 1Department of Animal Science and Technology, National Taiwan University, Taipei 10617, Taiwan; r07626015@ntu.edu.tw (K.-Y.L.); yaoyao@ntu.edu.tw (S.-Y.W.); 2Department of Biochemistry and Molecular Biology, National Yang Ming Chiao Tung University, Taipei 112304, Taiwan; tsaiyc@ym.edu.tw; 3Department of Animal Science, National Chung Hsing University, Taichung 40227, Taiwan; chenyp@dragon.nchu.edu.tw

**Keywords:** *Lactobacillus paracasei* PS23, fermented milk, coculture strategy, intestinal barrier protection

## Abstract

Few studies have documented the effects of fermented milk on intestinal colitis, which are mediated by regulating various microbial and inflammatory processes. Here, we investigated the effects of fermented milk with *Lactobacillus paracasei* PS23 on intestinal epithelial cells in vitro and dextran sulfate sodium (DSS)-induced colitis in vivo. As *L. paracasei* PS23 grew poorly in milk, a coculture strategy with yogurt culture was provided to produce fermented milk (FM). The results indicated that the coculture exhibited a symbiotic effect, contributing to the better microbial and physicochemical property of the fermented milk products. We further evaluated the anti-colitis effect of fermented milk with *L. paracasei* PS23 in vitro. Both PS23-fermented milk (PS23 FM) and its heat-killed counterpart (HK PS23 FM) could protect or reverse the increased epithelial permeability by strengthening the epithelial barrier function in vitro by increasing transepithelial electrical resistance (TEER). In vivo analysis of the regulation of intestinal physiology demonstrated that low-dose *L. paracasei* PS23-fermented ameliorated DSS-induced colitis, with a significant attenuation of the bleeding score and reduction of fecal calprotectin levels. This anti-colitis effect may be exerted by deactivating the inflammatory cascade and strengthening the tight junction through the modification of specific cecal bacteria and upregulation of short-chain fatty acids. Our findings can clarify the role of *L. paracasei* PS23 in FM products when cocultured with yogurt culture and can elucidate the mechanisms of the anti-colitis effect of *L. paracasei* PS23 FM, which may be considered for therapeutic intervention.

## 1. Introduction

Mucosal barriers constructed by intestinal epithelial cells, segregating gut microbes and the host immune system, play a crucial role in intestinal homeostasis to prevent commensal microbes or invading pathogenic microorganisms [1]. Different factors affect the immune system at the mucosal level by producing mediators, including cytokines and chemokines, to induce T-cell immune responses [2], and activated B cells differentiate into IgA plasma cells [3]. As the intestinal barrier function can be regulated by intestinal microbes through various mechanisms [4], probiotics can be used therapeutically to regulate the intestinal barrier function in intestinal inflammation-related diseases [5]. Numerous clinical trials have demonstrated that probiotics ameliorate intestinal diseases, including ulcerative colitis, irritable bowel syndrome, *Clostridium difficile* infection, antibiotic-associated diarrhea, and necrotizing enterocolitis [6]. Probiotics can increase the ability of the intestinal epithelial barrier to inhibit pathogenic bacterial attachment to the gut mucosa [7] by upregulating the expression of defensins, mucins [8], and proteins related to tight junctions [9] and by regulating immune responses through regulatory T cells and dendritic cell (DC) [10]. However, few studies have documented the effects of fermented milk (FM) on intestinal colitis, which are mediated by regulating various microbial and inflammatory processes at the intestinal level [11].

*L. paracasei* PS23 (PS23), isolated from healthy human feces could delay age-related cognitive decline, improve anxiety-like behaviors, and increase serotonin and dopamine levels in both the hippocampus and striatum in senescence-accelerated mouse prone 8 mice by preventing oxidation and inflammation and modulating gut–brain axis communication [12]. In maternal separation mice, the administration of live and heat-killed PS23 improved anxiety- and depression-like behaviors and decreased serum corticosterone levels, which were accompanied by higher serum anti-inflammatory interleukin (IL)-10 levels [13]. Loss of the gut–blood barrier and a disrupted intestinal barrier have been reported to associate with various neurological diseases, including Alzheimer’s disease and depression [14].

Adding probiotic strains to food products, especially FM products, is an efficient approach to consume beneficial microbes. Although the consumption of FM with probiotics considerably improves health, most probiotic functional studies have only focused on probiotics without considering the effect of the milk matrix and starter cultures. Several factors affect the viability and functionalities of probiotics, including matrix acidity, level of oxygen in products, presence of other lactic acid bacteria, and sensitivity to metabolites produced by other competing bacteria [15].

The development of new functional fermented dairy products, such as FM with probiotics, can provide major health benefits. Thus, we investigated the effects of FM with *L. paracasei* PS23 on intestinal epithelial cells (Caco2-C2BBe1) in vitro and DSS-induced colitis in vivo. We also evaluated the possible mechanisms underlying the protective effects of PS23, including cytokine products and intestinal barrier restoration.

## 2. Materials and Methods

### 2.1. Preparation of L. paracasei PS23

*L. paracasei* PS23, previously isolated from infant feces and deposited at the Leibniz Institut DSMZ-Deutsche Sammlung von Mikroorganismen und Zellkulturen GmbH with accession number DSM 32322, was cultured in de Man, Rogosa, and Sharpe (MRS) broth (Difco Laboratories, Detroit, MI, USA) at 37 °C and harvested in the log phase by washing and resuspending 3 times in PBS (Hyclone, South Logan, UT, USA). After washing, lactic acid bacteria (LAB) cells were resuspended in PBS, and the suspension was adjusted to the desired concentrations.

### 2.2. Preparation of FM with PS23

Skim milk powder was homogenized in distilled water (11 g/100 mL), pasteurized at 85 °C for 30 min, and cooled to room temperature. The milk was then inoculated with 4% yogurt culture (*L. delbruckii* subsp. *bulgaricus* and *Straptococcus thermophilius*) with 0.01% or 0.1% PS23 powder (5 × 10^10^ CFU/g). The inoculated milk was incubated at 37 °C for 6 h and then stored at 4 °C. The media of MRS-V, MRS agar pH 5.2 (MRS 5.2), and M17 (M17 agar, LAB092, Neogen, Scotland, UK) were used to evaluate the counts of *L. paracasei*, *L. delbruckii* subsp. *bulgaricus*, and *Streptococcus thermophilus*, respectively [16]. The heat-killed PS23 FM was prepared by heating PS23 FM at 95 °C for 40 min and inactivation was confirmed by the absence of colony formation on the MRS agar plate.

### 2.3. Determination of Physicochemical Properties of FM

Lab 850 Benchtop pH Meters (Mainz, Germany) were used to detect the pH of fermented milk FM products. The syneresis and concentration of EPS were determined as previously described. [17] A Rheometer (RST-CPS Touch Rheometer, Ametek Brookfield, MA, USA) was used to determine the apparent viscosity of FM at a shear velocity of 25 rpm. The concentrations of acetic acid and lactic acid were determined on an Agilent1200 HPLC system, as described by Wang et al. [18].

### 2.4. Caco-2 Epithelial Monolayer

The human colonic epithelial cell line Caco2-C2BBe1 was obtained from the American Type Culture Collection (Manassas, VA, USA) and cultured in Dulbecco’s modified Eagle’s medium supplemented with 10% heat-inactivated fetal bovine serum, 50 μg/mL penicillin, 50 μg/mL streptomycin sulfate, and 100 μg/mL neomycin sulfate (Invitrogen, Carlsbad, CA, USA). An intestinal epithelial monolayer was formed as per the method of Chen et al. [19]. Briefly, the cells were cultured under a humidified atmosphere of 5% CO_2_ at 37 °C. The Caco-2 cells were seeded onto permeable 12-well Transwell membranes (Corning, Lowell, MA) with a 3-μm pore size (density, 10^5^/cm^2^) to form the intestinal epithelial monolayer. The culture medium was replaced with fresh medium every 2 days during the 28-day culture period. After 28 days, the transepithelial electrical resistance (TEER) of Caco-2 epithelial monolayers was >300 Ω·cm^2^, indicating that the monolayers were ready to use [20]. The TEER was measured using an epithelial volt-ohmmeter with an STX2 probe (World Precision Instruments, Sarasota, FL, USA).

### 2.5. Effect of PS23 and PS23 FM on the Integrity of Caco-2 Epithelial Monolayer

For preparing 10^7^ CFU/mL PS23 FM, the milk was inoculated with 4% yogurt culture (*L. delbruckii* subsp. *bulgaricus* and *Straptococcus thermophilius*) with 0.02% PS23 powder (5 × 10^10^ CFU/g). The inoculated milk was incubated at 37 °C for 6 h and then stored at 4 °C. Caco-2 monolayers were cocultured with 10^7^ CFU/mL of PS23 or PS23 FM at 37 °C for 24 h, followed by the addition of 3% Dextran sulfate sodium (DSS, 36,000–50,000 M.W., CAS 160,110, MP, Biomedicals, France) and incubation for 30 h [21]. The TEER values before coculture with samples and after treatment with DSS were measured at various time intervals. The TEER was calculated as the ratio of the TEER at time t to the initial value for each series. The TEER of monolayers without added bacteria represented the controls for each experiment.

### 2.6. DSS-Induced Colitis Animal Model

The animal study was approved by the Animal Ethics Committee of National Taiwan University (NTU 2019-0095). Specific-pathogen-free 7-week-old female C57BL/6 mice, purchased from the National Laboratory Animal Center (Taipei, Taiwan), were housed in the Animal Experiment Center of National Taiwan University under standard conditions: 25 ± 2 °C, 55% ± 5% humidity, and 12 h light–dark cycle. All animals (48 mice) were fed a standard pelleted diet and sterilized water. The DSS-induced colitis model in mice was modified from that described by Wirtz et al. (2007) [22]. Mice with similar bodyweight were separated into groups of 8, and each group was administered 0.2 mL of FM, FM containing 10^7^ CFU PS23 (low-dose PS23 FM group, L PS23 FM), heat-killed low-dose PS23 FM (HKL PS23 FM), FM containing 10^8^ CFU PS23 (high-dose PS23 FM group, H PS23 FM), or heat-killed high-dose PS23 FM (HKH PS23 FM) intragastrically daily for 14 days. During the last 7 days of administration, 2% DSS (molecular weight: 36,000–50,000 Da; MP Biomedicals, Aurora, OH, USA) was added to the drinking water to induce colitis. The 2% DSS solution was replaced with fresh DSS every 2 days. The mice were anesthetized with isoflurane (Abbott Laboratories, Kent, UK) and then killed by cervical dislocation. The colon was removed, and the length of the colon was measured. The feces in the colon were detached by flushing with ice-cold PBS. A 1-cm-long fragment from the distal part of the colon was excised, washed with ice-cold PBS, and immediately immersed in 10% histological-grade phosphate-buffered formalin (Mallinckrodt Chemical, Derbyshire, UK).

### 2.7. Assessment of Intestinal Bleeding and Determination of Fecal Calprotectin

For the collection of feces, mice were removed from their cages and placed individually in circular plastic containers (15 cm in diameter × 18 cm in height) with filter paper on the bottom. Fecal pellets were collected (approximately 150–200 mg of feces per sample) after 1 h of isolation. The intestinal bleeding score system (Wirtz et al., 2007) [22] was used to assess stool consistency and bleeding. Occult blood in the feces was measured using a Hemoccult Sensa (Beckman Coulter, Brea, CA, USA). Calprotectin was analyzed using the enzyme-linked immunosorbent assay (ELISA) [23].

### 2.8. Histological Evaluation

Fixed colon tissue samples were dehydrated in ethanol and further embedded in paraffin wax, sectioned (5-μm thickness), and stained with hematoxylin–eosin. The histological score was evaluated by well-trained histologists according to the method of Wirtz et al. [22].

### 2.9. Colon Organ Culture and Cytokine Production

Colonic cytokine production was measured using an ex vivo colon organ culture, which was described by Gibson et al. [24]. The supernatants were collected by centrifugation at 13,400× *g* and 4 °C for 3 min to remove tissue debris and were then stored at −80 °C until ELISA. The cytokine levels in the supernatants were measured using DuoSet ELISA development systems (R&D Systems, Minneapolis, MN, USA) according to the manufacturer’s instructions.

### 2.10. Determination of Occludin and Myeloperoxidase

Intestinal epithelial cells were lysed in buffer containing 0.1 M Tris-HCl (pH 7.5)/2% SDS/10% glycerol/5% 2-mercaptoethanol, boiled at 95 °C for 5 min, centrifuged at 15,000 rpm for 5 min, and analyzed through reducing SDS-PAGE. Immunoblotting was performed using antibodies against occludin (Abcam, Cambridge, UK, ab15098, 1:100), myeloperoxidase (MPO, Abcam), and β-actin (Santa Cruz, TX, USA, sc-1615, 1:200). Quantitative analysis of Western blotting was performed using the Scion Image 4.0 (Scion Corporation, Frederick, MD, USA) and relative intensities of the target proteins to β-actin were shown.

### 2.11. Determination of Specific Cecal Bacteria

DNA was extracted from cecal contents. Enterobacteriaceae, *Bifidobacterium* genus, and *C. perfringens* were quantified through quantitative polymerase chain reaction (qPCR) using specific primers according to the procedures of Lubbs et al. [24] and Krych et al. [25].

### 2.12. Statistical Analysis

The data are expressed as mean ± standard deviation. All results were analyzed using the ANOVA GLM procedure in Statistical Analysis Systems v9.4 (SAS Institute Inc., Cary, NC, USA). Comparisons among multiple groups were processed with Tukey’s HSD (honest significant difference) test. *p* < 0.05 was set to indicate a significant difference. All experiments were performed in triplicates.

## 3. Results

### 3.1. Coculture with PS23 Improved the Microbial and Physicochemical Properties of FM Products

In our preliminary test, *L. paracasei* PS23 could not acidify milk to form a typical FM texture due to the inability to use lactose as a carbon source (data not shown). The coculture strategy with yogurt culture (*L. delbruckii* subsp. *bulgaricus* and *S. thermophilius*) was then implemented, which had a great effect on the acidification of the samples. The pH and titratable acidity (TA) of all three samples with yogurt culture (FM, FM+0.01 PS23, FM+0.1 PS23) decreased and increased to 4.6 and 0.6%, respectively, within 8-h fermentation (Figure 1A). We also noticed that coculturing with PS23 shortened the fermentation time of FM and accelerated the growth rate of yogurt cultures. Compared with the yogurt culture counterpart (FM group), the FM+0.01 PS23 and FM+0.1 PS23 groups had significantly lower pH and higher TA at 6 h fermentation and had significantly higher bacterial counts of *L. delbruckii* subsp. *bulgaricus* and *S. thermophilius* at 4 and 6 h fermentation (All *p* < 0.05). However, the coculture strategy did not improve LP23 growth in milk. The LP23 numbers after 8 h fermentation in both FM+0.01 PS23 and FM+0.1 PS23 groups were 7.54 and 8.41 Log CFU/mL, respectively, which only slightly increased from the original inoculating bacterial numbers (Figure 1B).

The physicochemical properties of the different FM products are presented in Figure 1C and Table 1. The addition of PS23 significantly and dose-dependently increased the firmness, consistency, cohesiveness, and viscosity and reduced the syneresis (*p* < 0.05). The EPS concentrations exhibited a similar trend as the results of viscosity and syneresis. The EPS levels in the FM+0.01PS23 and FM+0.1PS23 groups were 0.18 and 0.25 g/L, respectively, which were significantly higher than that in the FM counterpart (*p* < 0.05).

Analysis of the microbial and physicochemical properties of FM products with *L. paracasei* PS23 during the 21-day storage (Figure 2) indicated that the pH value and TA of the three groups were gradually decreased and increased, respectively, during storage without significant between-group differences (Figure 2A). For microorganisms (Figure 2B), *L. delbruckii* subsp. *bulgaricus* decreased during the storage period, and this was more pronounced after 7 days of storage. At the end of the storage, the numbers of *L. delbruckii* subsp. *bulgaricus* in the FM, FM+0.01PS23, and FM+0.1PS23 groups were 7.23, 6.93, and 6.99 Log CFU/mL, respectively, whereas no change was observed for *S. thermophilus*, which was maintained at approximately 9 Log CFU/mL. PS23 viability was also stable in both FM+0.01% PS23 and FM+0.1% PS23 groups during storage (7.86 and 8.46 Log CFU/mL, respectively). The sensory evaluation test indicated no difference among the three groups in terms of appearance, order, texture, and flavor (data not shown).

### 3.2. PS23 FM Enhanced Intestinal Epithelial Barrier Function In Vitro

The in vitro intestinal epithelial barrier function was investigated using the polarized intestinal epithelial monolayer formed by the Caco-2 cell line. PS23 FM (10^7^ CFU/mL) and heat-killed PS23 FM (HK PS23 FM) significantly increased the TEER of polarized Caco-2 monolayers (*p* < 0.05) compared with milk, FM, and PS23 powder (10^7^ CFU/mL) before DSS treatment (*p* < 0.05) (Figure 3A,B). We further assessed the samples for their protective effects on the intestinal epithelial monolayer against DSS challenge. The results indicated that the DSS challenge significantly damaged intestinal epithelial cells (Figure 3A). However, after a 6 h DSS challenge (30 h), the TEER value of the HK PS23 FM group was significantly higher than that of the other groups (Figure 3A,B).

### 3.3. Low-Dose PS23 FM Ameliorated DSS-Induced Colitis In Vivo

The anti-colitis potential of FM with *L. paracasei* PS23 in vivo was subsequently assessed. Compared with the colitis control mice, the mice administered low-dose *L. paracasei* PS23 FM (L PS23 FM) and the heat-killed counterpart (LHK PS23 FM) showed a significantly lower rectal bleeding score (Figure 4A). The fecal calprotectin result was consistent with fecal bleeding. The mice in the DSS colitis group had significantly upregulated fecal calprotectin compared to the FM control mice, and fecal calprotectin was downregulated in the HHK PS23 FM and LPS23 FM groups (Figure 4A). Additionally, reduction in the colon length in DSS colitis mice was non-significantly attenuated in both HHK PS23 FM and LPS23 FM groups (Figure 4B). Histological analysis indicated that the colitis control animals displayed signs of severe colitis, with a high degree of inflammatory infiltrates in the colonic mucosa, loss of goblet cells, and a disturbed mucosal architecture. However, inflammation was reduced in mice treated with low-dose PS23 FM, as reflected by rectum histologic score without significant difference (Figure 4C).

### 3.4. Low-Dose PS23 FM Regulated Intestinal Tight Junction Protein, Cytokine Secretion in Mesenteric Lymph Modes, Cecal Luminal Bacteria, and Cecal Short-Chain Fatty Acid

The results of tight junction proteins supported the ameliorating ability of HHK PS23 FM and LPS23 FM. Expression of occludin and MPO was significantly decreased and increased, respectively, in DSS colitis mice (Figure 5A). The administration of FM cocultured with PS23 upregulated occludin levels. This effect became statistically significant in the mice fed with heat-killed high-dose PS23 FM (*p* < 0.05). For MPO expression, both the HHK PS23 FM and LPS23 FM groups showed significantly downregulated occludin expression compared with the DSS group (*p* < 0.05).

The cytokine production profile in mesenteric lymph nodes (MLN) indicated that secretion of the inflammatory cytokine (IL-22) and anti-inflammatory/regulatory cytokine (IL-10) was significantly upregulated and downregulated, respectively, in DSS colitis mice (Figure 5B) compared with control mice. The administration of PS23 FM, except for HLK PS23 FM, reduced and increased the production of IL-22 and IL-10, respectively, induced by DSS (Figure 5B).

Specific cecal bacteria analysis through qPCR indicated that the numbers of *Clostridium* spp. and *Escherichia* spp. significantly increased in DSS colitis mice (Figure 5C) compared with control mice. Both HHK PS23 FM and LPS23 FM groups showed nonsignificantly reduced *Clostridium* spp. and *Escherichia* spp. numbers. Short-chain fatty acid analysis revealed that the administration of low-dose PS23 FM significantly increased butyric acid production compared with that in the DSS group (Figure 5D).

## 4. Discussion

A minimum amount of living probiotics is required for a positive effect of their consumption [26]. In addition, the rate of acid development is a critical factor in milk fermentation for preventing the growth of undesirable microorganisms and for ensuring the aroma, texture, and flavor of the end-product [27]. *L. paracasei* PS23 grew poorly in milk due to the inability to utilize lactose. Similar results were also reported for the strain *L. paracasei* F19 with a fermentation time of >20 h required to reach pH 4.5 [28]. The development of symbiotic combinations is another approach to developing functional food to stimulate probiotic growth. The present study provides the first coculture strategy to prepare a yogurt-like product with *L. paracasei* PS23. This strategy successfully accelerated acid production by stimulating the growth of both yogurt cultures after 4 and 6 h fermentation. Kristo et al. [29] emphasized the importance of inoculating yogurt culture (*Lactobacillus delbrueckii* subsp. *bulgaricus* and *S. thermophilus*) for the acidification rate for FM with probiotics at the beginning. *L. paracasei* possesses proteolytic activity to stimulate the growth of other bacteria [27]. However, the coculture strategy did not improve the growth of *L. paracasei* PS23 in the FM samples.

For FM products, the probiotic culture must contribute to the good texture and sensory properties of the final product [30]. Our coculture approach with PS23 increased the viscosity and prevented the syneresis of the FM products, which were paralleled with EPS concentrations without affecting the sensory properties. EPS-producing LAB has been commonly used in the dairy industry to improve the viscosity and texture of the milk products. Growth conditions, such as culture media, temperature, and pH, can affect EPS production by LAB, which further affects both technological and probiotic properties [31]. Growth under suboptimal conditions reduces the biosynthesis of bacterial cell wall polymers, leading to the increased availability of isoprenoid lipid carrier precursor molecules [32] and the increased activities of enzymes involved in the synthesis of precursors for EPS biosynthesis [33].

Although the coculture strategy did not stimulate the growth of *L. paracasei* PS23 during fermentation, the level of inoculum affected the number of bacteria at the end of fermentation and subsequently affected the number during storage. The levels of *L. paracasei* PS23 were maintained during 8 h fermentation and 21-day storage, suggesting the good compatibility of the PS23 strain with the yogurt culture (*L. delbrueckii* subsp. *bulgaricus* and *S. thermophilus*) and good viability in the acidic environment of the FM product. A previous study also found a high level of *L. paracasei* B 117 during storage with the supporting culture, which indicates the acid tolerance of this strain under the acidic environment of the FM product [29]. The findings suggested that coculturing PS23 with yogurt culture exerted a symbiotic effect, contributing to the better microbial and physicochemical properties of the FM products.

We evaluated the anti-colitis effect of FM with *L. paracasei* PS23 in vitro. Studies on the intestinal protective effects of *L. paracasei* have mostly focused on immunomodulation and antibacterial activity [34,35,36]. The current study is one of few papers directly investigating the effects of milk product fermented with *L. paracasei* on intestinal epithelial cells. Both PS23 FM and its heat-killed counterpart (HK PS23 FM) demonstrated good potential to protect or reverse the increased permeability of the epithelium by strengthening the epithelial barrier function in vitro through increases in the TEER. This approach is a sensitive method for determining permeability in vitro [20] and has been widely applied for screening probiotics and functional compounds with intestinal protective effects [19]. Notably, only FM samples with PS23 (live and heat-killed)—and not the milk matrix, FM, and PS23 strain—exerted intestinal-protecting effects, implying the involvement of the metabolites/bioactive compounds or probiotic DNA/cell wall of *L. paracasei* PS23 in PS23 FM. Our findings agree with those of Thoreux et al. [37] and Chen et al. [11], demonstrating that the supernatant of *L. paracasei* DN114001-fermented low-fat milk and heat-killed *L. paracasei* 01 FM can increase the proliferation of intestinal epithelial cells.

Further in vivo effects on the regulation of intestinal physiology demonstrated that FM with low-dose *L. paracasei* PS23 (LPS23 FM) ameliorated DSS-induced colitis, with a significant attenuation in the bleeding score and reduction of fecal calprotectin. A prominent feature of mucosal histology in patients with active inflammatory bowel disease (IBD) is infiltration by neutrophilic granulocytes [38]. Fecal calprotectin, a major protein of neutrophils, is a fecal marker of IBD severity [39] and has a strong positive correlation with the fecal excretion of 111 indium-labeled neutrophils [40]. The decrease in fecal calprotectin levels in the LPS23 FM group represented diminishing mucosal neutrophil infiltration, which was associated with a reduction in fecal blood loss, a major clinical sign of IBD.

We investigated the possible mechanism through which low-dose PS23 FM diminished DSS-induced symptoms. Colitis is characterized by the recruitment of circulating leukocytes in gut tissues and the release of proinflammatory mediators [38]. In mice receiving PS23 FM, the inflammatory cascade induced by DSS was deactivated. First, MPO expression in colon, an index of leukocyte infiltration, decreased significantly in the LPS23 group, suggesting decreased leukocyte recruitment. Second, fecal calprotectin was diminished, indicating that mucosal neutrophil infiltration was alleviated. Third, the proinflammatory cytokine (IL 22) and anti-inflammatory cytokine (IL10) were regulated in the colon, demonstrating an improvement of inflammatory responses. In most acute inflammatory responses, IL-10 can mediate a compensatory anti-inflammatory response to alleviate systemic inflammatory response syndrome [41]. Conversely, IL-22, mainly produced by T helper 17 (Th17) cells and Th22 cells, can be induced under inflammatory conditions, such as IBD. The clinical relevance of IL-22 to IBD has been highlighted. Mizoguchi et al. [42] indicated that IL-22 expression, which is elicited by inflammatory insults, is further affected by local cytokine environments, such as Th1 or Th2 response. A dominant Th2 response causes lower IL-22 levels, consistent with our finding: PS23 FM upregulated the level of IL10 in the DSS-induced colitis mouse model to create a more anti-inflammatory environment, which inhibited IL22 expression.

In addition to the anti-inflammatory effect, strengthening the tight junction might be another mechanism involved in the anti-colitis effect of PS23 FM through increases in occludin synthesis and short-chain fatty acid production. Epithelial barrier function is regulated by tight junctions [43]. Tight junctions comprise transmembrane proteins, such as occludin, which is a 65-kDa integral plasma-membrane protein. Intestinal permeability disorders markedly decrease its expression. Patients with Crohn’s disease have low occludin levels and severely compromised tight junction integrity [44]. Short-chain fatty acids promote epithelial barrier function and integrity by reinforcing tight junctions [45], and they regulate inflammasome-mediated inflammatory responses [46]. A recent study indicated that butyrate maintained and/or promoted the TEER through the induction of genes encoding tight junction components and protein reassembly [18]. Moreover, short-chain fatty acids regulate epithelium/luminal bacteria interaction by stimulating the production of antimicrobial peptides by intestinal epithelial cells as the first-line defense against harmful bacteria. This finding is consistent with our specific cecal bacteria results. The administration of PS23 FM reduced the levels of harmful bacteria, *Clostridium* spp. and *Escherichia* spp., both of which are involved in the incidence and severity of IBD. *C. difficile* colonized the large bowel of patients undergoing antibiotic therapy and secreted two toxins (TcdA and TcdB), causing disease pathologies [47]. *E. coli* flagellin in the leaking mucosal barrier could pass through the tight junctions, leading to IL-8 release from colon epithelial cells by flagellin-Toll receptor 5 (TLR5) interaction [48]. A higher number of mucosa-associated *E. coli* was observed in patients with ulcerative colitis. The mechanism through which short-chain fatty acids inhibit bacterial growth is suggested to involve intracellular acidification of bacterial cells, leading to decreased transmembrane potentials and disturbed cellular biological activities [49]. Fach et al. [50] also demonstrated that butyrate can protect intestinal epithelial cells from damage caused by *C. difficile* toxins by activating the transcription factor HIF-1 and regulating local inflammation.

## 5. Conclusions

In the present study, we successfully prepared FM through coculture with the novel strain *L. paracasei* PS23, which provided better microbial and physicochemical properties than FM without PS23 through a symbiotic effect. The results of our in vitro and in vivo colitis studies revealed that low-dose *L. paracasei* PS23 FM diminished DSS-induced symptoms by deactivating the inflammatory cascade and strengthening the tight junction by modifying the specific cecal bacteria and upregulating short-chain fatty acids. Heat-killed PS23 also exerted a positive effect on colitis, indicating that the metabolites/bioactive compounds or probiotic DNA/cell wall of *L. paracasei* PS23 in the FM might provide protective effects on intestinal cells. Thus, our findings not only clarify the role of *L. paracasei* PS23 in FM products when cocultured with yogurt culture but also elucidate the mechanisms of the anti-colitis effect of *L. paracasei* PS23 FM, indicating an opportunity for therapeutic intervention.

## Figures and Tables

**Figure 1 foods-10-02337-f001:**
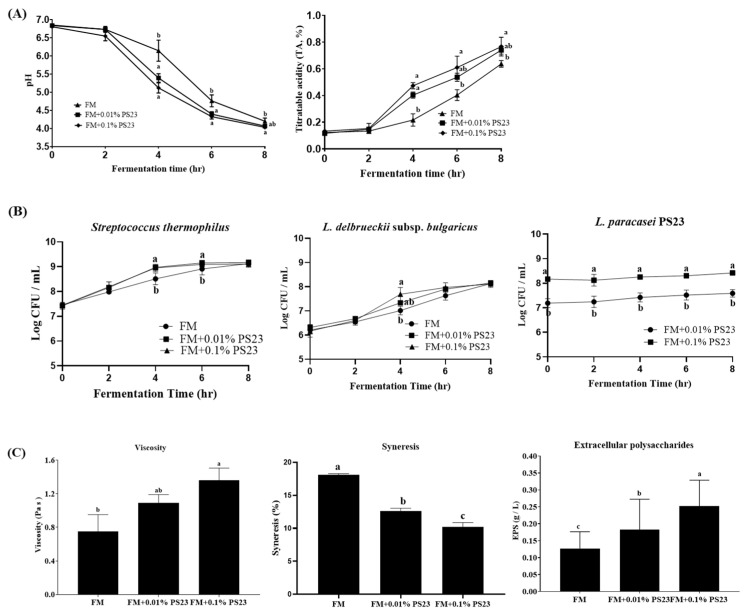
Effect of coculturing PS23 with yogurt culture (*Lactobacillus delbrueckii* subsp. *bulgaricus* and *Streptococcus thermophilus*) on (**A**) acidification, (**B**) bacterial counts, and (**C**) rheology of the fermented milk. Data are expressed as mean ± SD. Experiments were performed in three inde-pendent replicates. Statistical analysis was performed using analysis of variance with the Tukey test. Different letters indicate a significant difference (*p* < 0.05). PS23: *Lactobacillus paracasei* PS23, FM: fermented milk with yogurt culture. SD: standard deviation.

**Figure 2 foods-10-02337-f002:**
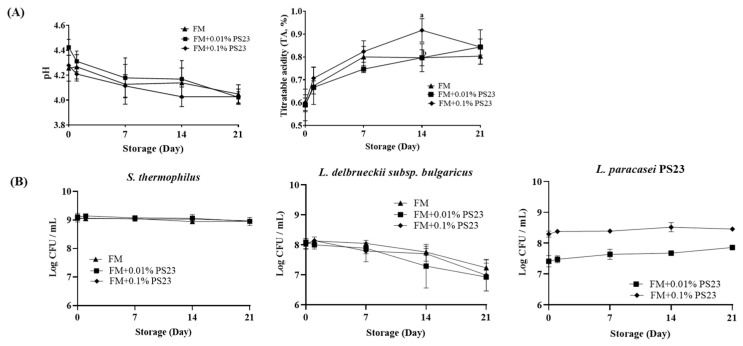
Effect of coculturing PS23 with yogurt culture (*Lactobacillus delbrueckii* subsp. *bulgaricus* and *Streptococcus thermophilus*) on (**A**) acidification and (**B**) bacterial counts of the fermented milk during 21-day storage. Data are expressed as mean ± SD. Experiments were performed in three independent replicates. Statistical analysis was performed using analysis of variance with the Tukey test. Different letters indicate a significant difference *p* < 0.05. PS23: *Lactobacillus paracasei* PS23, FM: fermented milk with yogurt culture. SD: standard deviation.

**Figure 3 foods-10-02337-f003:**
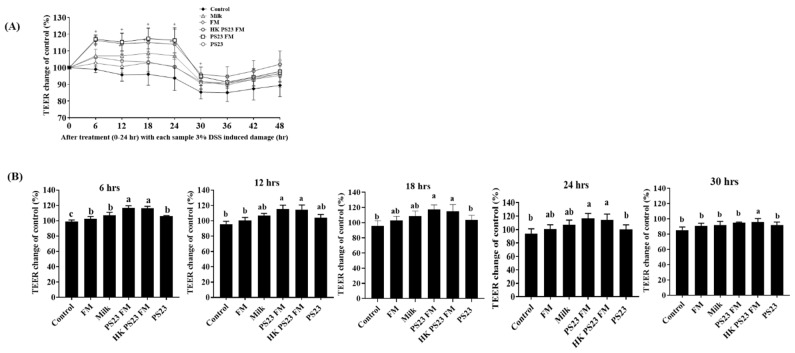
Effects of PS23-fermented milk on transepithelial electrical resistance (TEER) of Caco-2 cells w/wo DSS induction. (**A**) 0–48 h, (**B**) 6, 12, 18, 24, and 30 h of coculture. Data are expressed as means ± SD. Experiments were performed in three independent replicates. Statistical analysis was performed using analysis of variance with the Tukey test. Different letters indicate a significant difference *p* < 0.05. Control: monolayers without added samples; FM: fermented milk with yogurt culture; HK: heat-killed; PS23: *L. paracasei* PS23 powder (10^7^ CFU/g); PS23 FM: PS23 FM (10^7^ CFU/mL). SD: standard deviation.

**Figure 4 foods-10-02337-f004:**
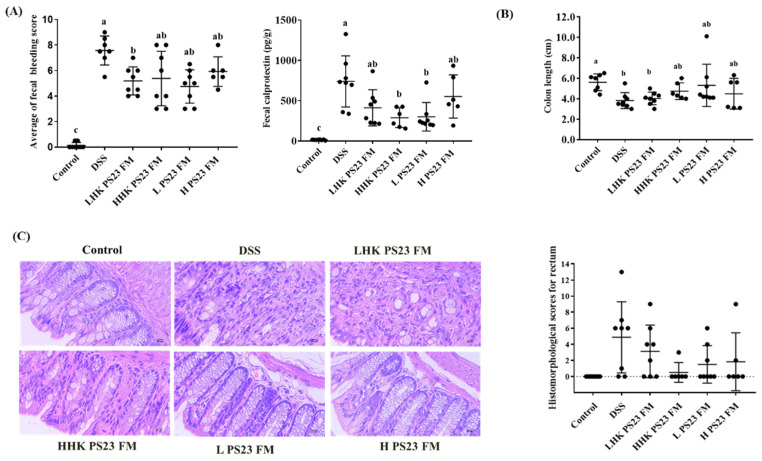
Effect of PS23-fermented milk on (**A**) fecal bleeding score and fecal calprotectin, (**B**) the lengths of colons, and (**C**) rectum histologic evaluation in a DSS-induced colitis mouse model. Data are expressed as means ± SD (*n* = 6–8). Statistical analysis was performed using analysis of variance with the Tukey test. Different letters in (**A**,**B**) indicate a significant difference (*p* < 0.05). DSS: negative control; LPS23 FM: low-dose *L. paracasei* PS23 with yogurt culture fermented milk (10^7^ CFU/mouse); LHK PS23 FM: heat-killed LPS23 FM; HPS23 FM: high-dose *L. paracasei* PS23 with yogurt culture fermented milk (10^8^ CFU/mouse); HHK PS23 FM: heat-killed HPS23 FM. SD: standard deviation.

**Figure 5 foods-10-02337-f005:**
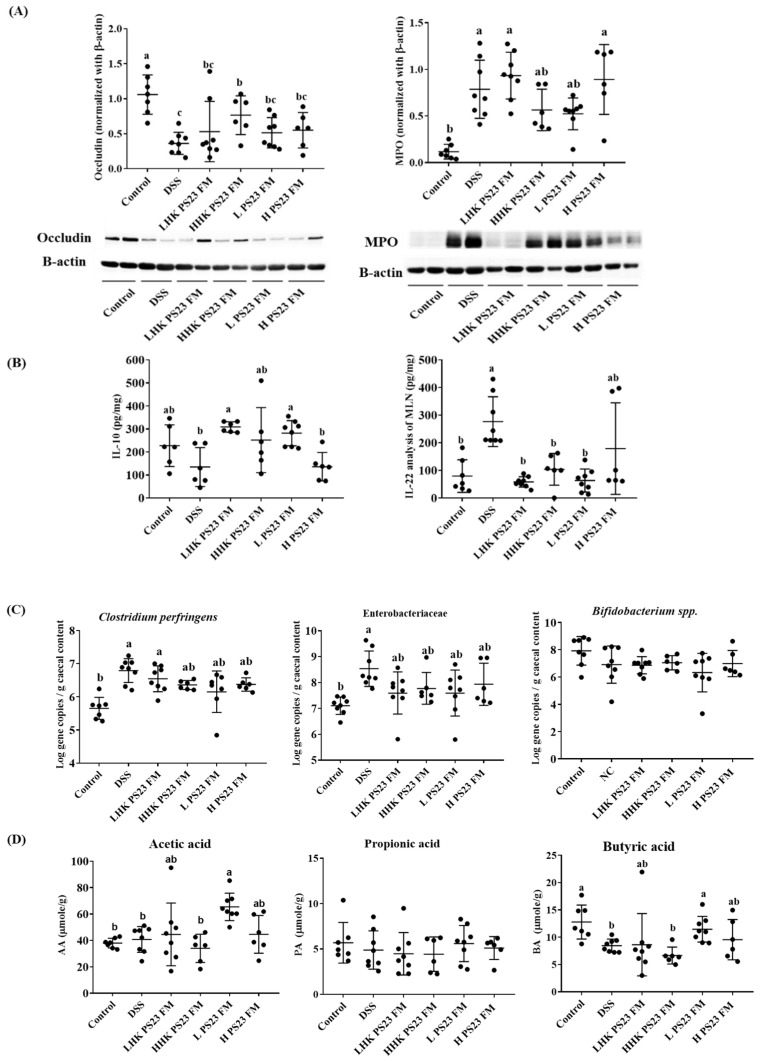
Effect of PS23-fermented milk on (**A**) tight junction protein expression in the large intestine, (**B**) cytokines in mesenteric lymph nodes (MLN), (**C**) cecal luminal bacteria, and (**D**) cecal short chain fatty acid in DSS-induced colitis mouse model. Data are expressed as means ± SD (*n* = 6–8). Statistical analysis was performed using analysis of variance with the Tukey test. Different letters indicate a significant difference *p* < 0.05. LPS23 FM: low-dose *L. paracasei* PS23 with yogurt culture fermented milk (10^7^ CFU/mouse); LHK PS23 FM: heat-killed LPS23 FM; HPS23 FM: high-dose *L. paracasei* PS23 with yogurt culture fermented milk (10^8^ CFU/mouse); HHK PS23 FM: heat-killed HPS23 FM. SD: standard deviation.

**Table 1 foods-10-02337-t001:** Textural properties of fermented milk samples with PS23.

	Textural Properties of Fermented Milk			
Groups	Firmness (g)	Consistency (g.s)	Cohesiveness (g)	Viscosity Index (g.s)
FM	191.52 ± 8.84 ^b^	4060.10 ± 155.42 ^b^	135.29 ± 8.42 ^c^	236.46 ± 12.96 ^c^
FM+0.01% PS23	202.51 ± 8.21 ^ab^	4296.43 ± 175.73 ^a^	179.32 ± 6.15 ^b^	283.04 ± 12.73 ^b^
FM+0.1% PS23	209.02 ± 5.26 ^a^	4390.41 ± 125.64 ^a^	219.97 ± 8.03 ^a^	349.65 ± 12.52 ^a^

^a^^–c^ Means with different letters in the same row column indicate a significant difference (*p* < 0.05).

## Data Availability

The data presented in this article is available on reasonable request, from the corresponding author.

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
