# Peer review of "Coculture Strategy for Developing Lactobacillus paracasei PS23 Fermented Milk with Anti-Colitis Effect"

_foods, 2021, doi:10.3390/foods10102337_

Round 1

Reviewer 1 Report

In this study, the authors investigated the in vitro and in vivo anti-colitis effect of Lactobacillus paracasei PS23 fermented milk. It’s an interesting study, however, some questions need to be considered.

Comments

  1. In the introduction, the authors stated that “The intestinal protective effect of PS23 might improve dysfunction of the blood-brain barrier, which warrants investigation.” However, how does its protective effect improve BBB was not discussed or studied. Please add more discussion in the text.
  2. The representative images of H&E to show the histological analysis need to be added. And why there are big error bars in fig.4C. If with the huge variation, how to prove the PS-23 fermented milk has an anti-colitis effect?   
  3. What are the abbreviations representing in fig.5A? What’s the duplicate indicate? Similar question, how to quantify the occludin expression with huge variation?
  4. Line 358, there is no test for MPO activity in the manuscript. The authors only showed decreased expression of MPO in the colon. 
  5. Figure 6 is a little confusing. Please simplify it. What are the Zn and Mn indicating? No discussion in the text
  6. The size and font of the text in the figures should be consistent and clear to see. Please revise them. 
  7. Some mistakes, for example, “107 CFU”, the x-axis of fig. 3B. 

Author Response

Dear Editors and reviewers:

Thank you for your email letter dated 09-24-2021 concerning the manuscript Foods-1388585 entitled “Co-culture Strategy for Developing Lactobacillus paracasei PS23 Fermented Milk with Anti-Colitis Effect”. According to the comments from the editors and reviewers, our initial manuscript has been carefully revised, and the point-by-point responses are listed below. We would appreciate it if our paper could be accepted for publication in Foods. I look forward to hearing from you soon.

Response to the reviewers

Reviewer 1

In this study, the authors investigated the in vitro and in vivo anti-colitis effect of Lactobacillus paracasei PS23 fermented milk. It’s an interesting study, however, some questions need to be considered.

Comments

  1. In the introduction, the authors stated that “The intestinal protective effect of PS23 might improve dysfunction of the blood-brain barrier, which warrants investigation.” However, how does its protective effect improve BBB was not discussed or studied. Please add more discussion in the text.

Response:

Thank you for the comments. We do not finish this part of study yet, thus, we delete “The intestinal protective effect of PS23 might improve dysfunction of the blood–brain barrier, which warrants investigation.”

  1. The representative images of H&E to show the histological analysis need to be added. And why there are big error bars in fig.4C. If with the huge variation, how to prove the PS-23 fermented milk has an anti-colitis effect?

Response:

Thank you for your suggestion. We have added the images of H&E in Figure 4 (Line 271 in the revised manuscript). We also changed the bar graph into scatter plot to show each data. Although the rectal histologic score did not show the significant difference due to the SD, we still can observe a higher number of mice from the PS23 groups demonstrating better histologic scores than those from the DSS group. Moreover, other data, such as fecal bleeding score, fecal calprotectin, and the lengths of colons, could also provide the evidences that administration of fermented milk with PS23 could ameliorate DSS-induced colitis in vivo.

Figure 4. Effect of PS23-fermented milk on (A) fecal bleeding score and fecal calprotectin, (B) the lengths of colons, and (C) colon histologic evaluation in a DSS-induced colitis mouse model. Data are expressed as means ± SD (n = 6-8). Statistical analysis was performed using analysis of variance with the Tukey test. P < 0.05. DSS: negative control; LPS23 FM: low-dose L. paracasei PS23 with yogurt culture fermented milk (107 CFU/mouse); LHK PS23 FM: heat-killed LPS23 FM; HPS23 FM: high-dose L. paracasei PS23 with yogurt culture fermented milk (108 CFU/mouse); HHK PS23 FM: heat-killed HPS23 FM. SD: standard deviation.

  1. What are the abbreviations representing in fig.5A? What’s the duplicate indicate? Similar question, how to quantify the occludin expression with huge variation?

Response:

Thank you for your correction. We have modified the abbreviation in Fig. 5A (Lines 300-301 in the revised manuscript). We changed the bar graph into spot to show each data. Although the SDs are high, the occluding levels still showed the significant difference among groups after ANOVA  and Tukey’s multiple range test.

  1. Line 358, there is no test for MPO activity in the manuscript. The authors only showed decreased expression of MPO in the colon.

Response:

Thank you for the correction. We have replaced “activity” with “expression in colon”. Please see Line 376 in the revised manuscript.

  1. Figure 6 is a little confusing. Please simplify it. What are the Zn and Mn indicating? No discussion in the text

Response:

Thank you for the suggestion. We have deleted this figure.

  1. The size and font of the text in the figures should be consistent and clear to see. Please revise them.

Response:

We have redrawn all the figures with consistent font and size. (Please see the figures in the revised manuscript).

  1. Some mistakes, for example, “107 CFU”, the x-axis of fig. 3B.

Response:

We have checked the whole manuscript thoroughly and corrected all the typos and mistakes.

Reviewer 2 Report

The authors have investigated the role of a potential probiotic, delivered with milk products, on the intestinal epithelium via different in vitro and in vivo assays. The paper is rather complete with different experiments though some modifications will be needed.

Methos:

Caco-2 epithelial monolayer

more details are needed here. People has to be able to replicate the experiment. For instance, with the information that you are giving, one has to assume that you are using trans well inserts but you do not say. Describe properly the experiment

Determination of microbiota

where did you measure microbiota? Fecal material of mice? Why you did not use 16S sequencing?

You should not use the term microbiota since you are only measuring three different bacteria. You can say something like 'determination of potentially harmful bacteria or implicated in inflammatory diseases'

Change the term throughout the manuscript

Results

PS23 FM Enhanced Intestinal Epithelial Barrier F unction I n V itro

you dont explain what the control is in TEER figures.
what exactly are the samples composed of: are samples diluted into cell growth medium? are the directly added to the cell line?
which molecular weight DSS?

figure 5D is not present

Art work:

Bar plots are discouraged since they hide data. There are several papers explaining why you should not use them and are easy to find with just typing "why not use barplots".

Instead, you can use dot plots which show all three points (three replicates)

Author Response

Dear Editors and reviewers:

Thank you for your email letter dated 09-24-2021 concerning the manuscript Foods-1388585 entitled “Co-culture Strategy for Developing Lactobacillus paracasei PS23 Fermented Milk with Anti-Colitis Effect”. According to the comments from the editors and reviewers, our initial manuscript has been carefully revised, and the point-by-point responses are listed below. We would appreciate it if our paper could be accepted for publication in Foods. I look forward to hearing from you soon.

Response to the reviewers

Reviewer 2

The authors have investigated the role of a potential probiotic, delivered with milk products, on the intestinal epithelium via different in vitro and in vivo assays. The paper is rather complete with different experiments though some modifications will be needed.

Methods:

  1. Caco-2 epithelial monolayer: more details are needed here. People has to be able to replicate the experiment. For instance, with the information that you are giving, one has to assume that you are using trans well inserts but you do not say. Describe properly the experiment.

Response:

Thank you for the suggestion. We have added the detail procedures regarding “formation of Caco-2 epithelial monolayer” as following: (Please see Lines 107-111 in the revised manuscript).

“The human colonic epithelial cell line Caco2-C2BBe1 was obtained from the American Type Culture Collection (Manassas, VA, USA) and cultured in Dulbecco’s modified Eagle’s medium supplemented with 10% heat-inactivated fetal bovine serum, 50 μg/mL penicillin, 50 μg/mL streptomycin sulfate, and 100 μg/mL neomycin sulfate (Invitrogen, Carlsbad, CA, USA). An intestinal epithelial monolayer was formed as per the method of Chen et al.[19]. Briefly, the cells were cultured under a humidified atmosphere of 5% CO2 at 37°C. The Caco-2 cells were seeded onto permeable 12-well Transwell membranes (Corning, Lowell, MA) with a 3-μm pore size (density, 105/cm2) to form the intestinal epithelial monolayer. The culture medium was replaced with fresh medium every 2 day during the 28-day culture period.”

  1. Determination of microbiota: where did you measure microbiota? Fecal material of mice? Why you did not use 16S sequencing?

Response:

We determined the specific bacteria from caecal contents. Since we focused on the specific gut bacteria, we did not perform next generation sequencing to investigate the gut microbiota.  (Please see Lines 176-177 in the revised manuscript).

“2.11. Determination of Specific Caecal Bacteria

DNA was extracted from caecal contents. Enterobacteriaceae , Bifidobacterium genus, and C. perfringens were quantified through quantitative polymerase chain reaction (qPCR) using specific primers according to the procedures of Lubbs et al.[24] and Krych et al.[25].”

  1. You should not use the term microbiota since you are only measuring three different bacteria. You can say something like 'determination of potentially harmful bacteria or implicated in inflammatory diseases' Change the term throughout the manuscript

Response: Thank you for your comments. We have replaced “microbiota” as “specific caecal bacteria” in the revised manuscript.

Results

  1. PS23 FM Enhanced Intestinal Epithelial Barrier F unction I n V itro you dont explain what the control is in TEER figures.

Response: The TEER of monolayers without added tested samples represented the control. The explanation has been added in the figure legend. (Please see Figure 3 in the revised manuscript, Lines 254-255)

Control: monolayers without added samples; FM: fermented milk with yogurt culture; HK: heat-killed; PS23: L. paracasei PS23 powder (107 CFU/g); PS23 FM: PS23 FM (107 CFU/mL).”

  1. what exactly are the samples composed of: are samples diluted into cell growth medium? are the directly added to the cell line?

Response:

The fermented milk samples were not diluted. For preparing 107 CFU/mL PS23 FM, the milk was inoculated with 4% yogurt culture (L. delbruckii subsp. bulgaricus and Straptococcus thermophilius) with 0.02% PS23 powder (5×1010 CFU/g). Preparation of 107 CFU/mL PS23 FM has been added to the revised manuscript (Lines 116-119).

  1. which molecular weight DSS?

Response:

The MW of DSS is 36,000-50,000 M.W. This information has been added in the revised manuscript (Lines 120-121).

“Caco-2 monolayers were cocultured with 107 CFU/mL of PS23 or PS23 FM at 37 °C for 24 h, followed by the addition of 3% Dextran sulfate sodium (DSS, 36,000-50,000 M.W., CAS 160110, MP, Biomedicals, France) and incubation for 30 h.”

  1. figure 5D is not present

Response:

Thank you for the correction. Figure 5D has been added in the revised manuscript (Line 301).

Art work:

  1. Bar plots are discouraged since they hide data. There are several papers explaining why you should not use them and are easy to find with just typing "why not use barplots". Instead, you can use dot plots which show all three points (three replicates)

Response:

Thank you for your comments. All the figures in animal study have been replaced by scatter plots according to your suggestion. Please see the Figure 4 and Figure 5 in the revised manuscript.

Round 2

Reviewer 1 Report

Thanks for the responses.